# CURSOR-BASED ADAPTIVE QUANTIZATION FOR DEEP NEURAL NETWORK

## ABSTRACT

Deep neural network (DNN) has rapidly found many applications in different scenarios. However, its large computational cost and memory consumption are barriers to computing restrained applications. DNN model quantization is a widely used method to reduce the DNN storage and computation burden by decreasing the bit width. In this paper, we propose a novel cursor based adaptive quantization method using differentiable architecture search (DAS). The multiple bits' quantization mechanism is formulated as a DAS process with a continuous cursor that represents the possible quantization bit. The cursor-based DAS adaptively searches for the desired quantization bit for each layer. The DAS process can be solved via an alternative approximate optimization process, which is designed for mixed quantization scheme of a DNN model. We further devise a new loss function in the search process to simultaneously optimize accuracy and parameter size of the model. In the quantization step, based on a new strategy, the closest two integers to the cursor are adopted as the bits to quantize the DNN together to reduce the quantization noise and avoid the local convergence problem. Comprehensive experiments on benchmark datasets show that our cursor based adaptive quantization approach can efficiently obtain lower size model with comparable or even better classification accuracy.

## 1 INTRODUCTION

Deep learning (DL) has achieved great successes in varied fields such as gaming, natural language processing, speech recognition, computer vision and so on. However, its huge computational burden and large memory consumption still intimidate many potential applications, especially for mobile devices and embedded systems.

A number of efforts have been devoted to compress the DL model size and accelerate its training and test speed. These efforts can be roughly categorized into four major classes: network pruning (Han et al. (2015); Anwar et al. (2015); Peng et al. (2019); Zhuang et al. (2018)), low rank approximation (Tai et al. (2015); Wang et al. (2018); Hayashi et al. (2019)), knowledge distillation (Hinton et al. (2015); Zagoruyko & Komodakis (2016)), and network quantization (Courbariaux & Bengio (2016); Lin et al. (2015); Wu et al. (2015); Polino et al. (2018); Zhang et al. (2018)). Among them, network quantization methods, jointly optimizing the whole network weights, activations or gradients with low bit (such as 8 bits or even 1 bit), show great potential in compressing model size and accelerating inference time. In addition, quantization based approaches are preferable for mobile devices and embedded systems since these devices are gradually equipped by specifically designed low bit computing hardware.

Although existing quantization based approaches, which mainly use one fixed bit to represent the whole DNN model, yields encouraging compression ratio while keeping the model's performance, we argue that simply using only a fixed bit for quantization is not the optimal choice for the trade-off between a model size and its performance. For example, to run a model on chips with strict memory limitations, 1 bit or 2 bits' quantization suffers from severe accuracy loss (Rastegari et al. (2016)) while 16 bits' or 8 bits' quantization cannot significantly reduce the model size.

To address the above problem, we propose a cursor based adaptive quantization method to derive a different number of bits in different layers for DNN model compression, i.e., we search for the best

configuration of different bit quantization for different layers in a neural network model. Distinctive from most algorithms aforementioned, our approach is motivated by recent neural architecture search (NAS) that aims to find better performance neural architecture with less calculations or less size automatically. The key in our algorithm is using a continuous cursor that represents the bit quantization scheme for each layer. For different layers, many cursors will be adaptively searched during the NAS process. Since the cursor itself is continuous and the whole search procedure can be considered as a differentiable architecture search (DAS) process, which can be effectively solved based on an alternative optimization strategy. A novel cost function that considers the model compression and prediction accuracy is also proposed in the DAS process. In the searching process of the cursors, a quantization process is applied to compress the model size at the same time. To reduce the possible quantization noise and local convergence problem, we make use of the closest two integer bits to the cursor to quantize the weights for each layer in a DNN model. We validate our proposed method with image classification tasks on CIFAR10,CIFAR100 and ImageNet. Comprehensive experiments on some backbone DNN models such as ResNet20, ResNet18, ResNet56 and MobileNetV2 show that the proposed cursor based quantization method achieves remarkably better performance of compression ratio with ignorable accuracy drop or even better accuracy.

In summary, the contributions of this work are four-fold:

- We cast the adaptive quantization of neural network as a problem of neural architecture search. A continuous cursor is proposed to represent the possible quantization bit, leading to a more efficient search space.
- A novel regularization function is proposed to optimize model compression in the search process. Thus the search for the cursor position and weights can be efficiently solved in an alternative optimization manner.
- Two nearest neighbor integers to the cursor are adopted with a carefully designed strategy to implement the quantization of the network to reduce the quantization noise and avoid possible local convergence.
- We comprehensively evaluate the proposed adaptive quantization method on some benchmark datasets and achieve new state-of-the-art performance for different number of bits quantization of neural network.

## 2 RELATED WORK

Recently, a lot of new quantization approaches have been proposed, enabling the quantized compressed model to compete with their full precision counterparts. Clustering method is applied for the weight codebook representation (Han et al. (2015)), and then the network is retrained to get better quantized centroids. In (Zhang et al. (2018)), the authors jointly trained a DNN and its associated quantizes to reduce the noticeable predication accuracy gap between the quantized model and its full precision one. A direct differentiable quantization method was introduced in (Louizos et al. (2018)) with promising test accuracy. A new activation quantization method that takes an activation clipping parameter was proposed in (Choi et al. (2018)) to ensure the suitable quantization scale.

Some efforts have been taken on quantization of the neural network with a different number of bits used for different layers. In (Lin et al. (2015)), signal-to-quantization-noise ratio (SQNR) is applied on layer weight to evaluate the effects of quantization error. Based on SQNR, different bits were used for quantization of each layer, yielding about 20% model size reduction without accuracy loss in their tests. The authors of (Wang et al. (2018)) proposed an automated mixed precision quantization scheme based on reinforcement learning (RL) to achieve better latency on different hardware platforms such as edge devices and cloud data center. They also claimed that their actor-critic model produced efficient actions that result in better latency and less energy consumption with negligible loss of accuracy.

In the past few years, a new trend has been witnessed for network design, i.e., neural architecture search (NAS). RL based approaches are first utilized to generate network with high accuracy (Zoph & Le (2016)), and they also build a strong basis for the following recent works such as (Gao et al. (2019); Guo et al. (2018)). Then, evolution based approach (Liang et al. (2018) is further applied to obtain the possible optimal solution in the large search space. Both of these two category approaches tend to yield large amount of computational burden because NAS is treated as a blackbox

optimization problem in a discrete domain, yielding a large number of architecture evaluations, and thus run very slow even on the most advanced GPU machine. To alleviate this problem, in 2018, the authors (Liu et al. (2018)) proposed a differentiable approach to accelerate the search of a good neural network by relaxation of the possible operation on the cell level structure. Wu et al. recently proposed a new approach to find the mixed bits for different layers by applying differentiable NAS (DNAS) method based on a model of super net (Wu et al. (2018)), which is a kind of directed acyclic graph. They considered the quantization as a problem of sampling on a stochastic super net, where a Gumbel softmax function is applied to make the sampling process differentiable.

We cast the different bit quantization for DNN as a cursor based adaptive architecture search problem, and it is different from the traditional direct quantization works and the learning based mixed bits' quantization approaches. Moreover, it is also distinctive from DARTs and DNAS in the methodology itself.

## 3 CURSOR-BASED ADAPTIVE QUANTIZATION

### 3.1 NEURAL ARCHITECTURE SEARCH

It is well known that DNN model needs much time to design its structure. Neural architecture search (NAS) recently emerged as a new methodology to overcome this challenge. It designs the optimal architecture of a neural network by considering all possible factors such as number of layers, width of each layer, different operators in each layer and so on. Two key concepts are related to a NAS process, i.e., search space and search strategy. All the possible combinations of the major factors that determine a network structure constitute the search space. Generally speaking, the search space of a DNN is very large, leading to a huge computational task. As such, the previous NAS works first devise normal and reduction cell (Pham et al. (2018)). Such kind of motif is then repeated to build the final neural network. Another definition is about search strategy, that is, how to transverse in such a large search space. With each searched network structure, the performance of it will be evaluated. A typical search method is random search, however, its computational efficiency is not ideal. Therefore, most recent works (Cai et al. (2018); Liu et al. (2018)) have been proposed along this big direction to improve the search efficiency as much as possible.

### 3.2 SEARCH SPACE FOR QUANTIZATION PROBLEM

Quantization has also been a very hot research topic in the past few years. Rounding function, vector quantization or stochastic function are typically applied to implement quantization to compact the model size while maintaining equivalent performance or acceptable loss. Some other approaches also use stochastic or probabilistic methods to quantize the neural network. Most previous methods simply apply one kind of bit quantization to the whole network due to the simplicity of implementation. A few recent works begin to utilize different bit quantization scheme to further improve the compression ratio and prediction accuracy. If we consider quantization choice as a part of the neural architecture, we can estimate its corresponding search space. Let us take Resent20 as an example and if we decide to quantize the neural network with the possible bit width of 1, 2, 4, 8, 16, 32, then all the possible quantization choices for ResNet20 will be $6^{20}$. In the context of NAS, this is a very large number for the search space. Hence, evaluation of so many designs one by one seems infeasible right now.

### 3.3 DIFFERENTIABLE CURSOR SEARCH FOR ADAPTIVE QUANTIZATION

The discrete search space of the above quantization scheme is so large. If we further consider the possible bit choice for each layer as a virtual continuous cursor in the range of $[1, 32]$, the cursors then become significant parts of the architecture for a neural network model. Here, we define the *cursor* as a position that is related to the quantization choice for each layer. Its value is a floating-point number within $[1, 32]$. If we assume a DNN has $N$ layers, and each layer has a different *cursor*, denoted by $c_1, c_2, ..., c_N$, together with their weights of $W_C$, our goal is to find a good combination of $c_1, c_2, ..., c_N$ to achieve better prediction accuracy and compression ratio. As such, for the whole neural network it can be described as an optimization problem that minimizes the loss on the validation data after training through the minimization of the loss on the training data as follows (Liu et al. (2018)):

$$\min E(x', y')_{D_T}(Loss_T(C, W_C)) \quad \text{s.t.} \quad W_{C*} = \text{argmin} E(x, y)_{D_V}(Loss_V(C, W_C)) \tag{1}$$

where $C$ represents the cursor vector, $W_{C*}$ is the weights corresponding to the optimal $C$, $Loss_T(C, W_C)$ and $Loss_V(C, W_C)$ is the respective training and validation loss function based on the cursors and weights with condition of $C$. $D_T$ and $D_V$ represents the training and validation dataset respectively, $(x, y)$ and $(x', y')$ means data from the training and validation dataset. It should be noted that using training and validation data is a tradition to derive the weight parameters and architecture parameters in the field of NAS, which is a little bit different from the other problems in deep learning. For simplicity and efficiency, in this paper, we let $Loss_T(C, W_C)$ equals to $Loss_V(C, W_C)$ and assume they share the same form of $Loss(C, W_C)$. To consider both the prediction accuracy and model size, we design the loss function as a combination of cross entropy and model size as follows:

$$Loss(C, W_C) = CrossEntropy(C, W_C) + \lambda \times Loss_C \tag{2}$$

where $CrossEntropy(C, W_C)$ is the widely used cross entropy function, encoding the prediction accuracy of the model. The reason why we add a regularization item to the loss function is first because it may determines the model's performance compromise between the accuracy and quantization, which directly determines the model size, and $\lambda$ is a regularization coefficient that adjusts the trade-off of accuracy and compression. In addition, it may prevent overfitting to some extent. Concerning the loss related to $Loss_C$, we focus on the model size change with quantization. So we design it in the form of Eq.(3), which will be introduced in details in the next subsection.

The above process is a bi-level optimization problem, which requires to deduce higher order derivatives and is hard to obtain an exact solution. An approximated iterative solution can be applied instead, so we alternatively take the optimization strategy in weight and cursor space to update $W_C$ based on the training losses from $D_T$ and renew $C$ based on the validation losses from $D_V$. By solving this bi-level optimization problem using an alternative approximation approach, the cursors can be efficiently searched by gradient based optimization approach such as Adam. Our later experimental results also show that this alternative optimization method may yield a good solution with high compression ratio and accuracy. Compared to the original discrete search space, this search method is more efficient because the design of continuous cursor and the direct gradient based optimization approach. The whole differentiable cursor search for adaptive quantization based on the alternative optimization of $W_C$ and $C$ is described in Algorithm 1. With the training and validation set, initialized cursor value and a pretrained model as inputs, our algorithm first quantizes the weights in each layer of the network using the two most close integers to the cursor and calculates the loss based on the training data for forward process, and then it updates the original 32 bit weight by gradient descent algorithm. For the subsequent validation, it also first quantizes the network with the two most close integers to the cursor, and use them to obtain the validation error, then our algorithm utilizes this error to update the cursors with gradient descend algorithm. It should be emphasized that we utilize the original 32 bit weight in the above training and validation step, and use it for sharing when implementing quantization with the cursor's two neighbor integers. The key step in the algorithm about quantization with the nearest two integers to the cursor will be elaborated in the subsequent section. The outputs of the whole algorithm are rounded cursor values for each layer and its quantized network model.

It should be noted that our proposed cursor based differentiable search is different from DARTs (Liu et al. (2018)) in the following three aspects. First, DARTs method considers the possible operation in each layer as a mixture of primitive operations. Here, we directly make use of cursor to represent the quantization bit for each layer, no similar mixture operation exists in the whole search algorithm. Second, in DARTs, each primitive operation is assigned with a probability through a softmax function. Cursor based search is optimized directly without probability. Third, DARTs approach concentrates on the cell structure, but we apply the DAS directly on the whole network. Compared to DNAS (Wu et al. (2018)), our approach is also distinctive. For DNAS, the authors build a stochastic super net first to describe all the possible quantization choices, then a sampling approach based on Gumbel softmax function that enables the discrete distribution to be continuous and differentiable is applied in each layer of the super net. Our cursor based differentiable search has no super net or sampling process in the pipeline. Hence, the subsequent solutions to the optimization problem is also completely different. In short, the proposed method requires no relaxation anymore as in both DARTs and DNAS approach.

---

**input** : The training set $D_T$, validation set $D_V$, initialized cursors $C$, pretrained 32-bit weight
$W$, and the batch size $n$

**while** *not reaching the target epochs or not converge* **do**

Sample data from training data $D_T$;

Quantize the network using two integers that are closest to the cursor, calculate the loss $L$
on training data with Eq.(2);

Update the weight $W$ by gradient descent $W' = W - \nabla_W L(C, W)$;

Sample data from validation data $D_V$;

Quantize the network using two integers that are closest to the cursor, calculate the loss $L$
on validation data with Eq. (2);

Update the cursor $C$ by gradient descent $C' = C - \nabla_C L(C, W)$;

**end**

**output:** Rounded cursor values for each layer and quantized network

---

**Algorithm 1:** Differentiable Cursor Search for Adaptive Quantization

### 3.4 TRAINING FOR NETWORK QUANTIZATION

Aiming for DNN quantization, we attempt to apply the cursor that represents the bit to quantize the
weight layers. Unfortunately, the cursor obtained during the search is a fractional number, which
cannot be directly used for quantization. One choice is to round the cursor to some integers, but it
may cause rather large quantization error if we choose the rather distant bits. On the other hand,
if we directly round the cursor to its nearest integer, it may not efficiently represent the variation
of cursor. For example, if $cursor1$ and $cursor2$ for different epochs in the same layer are 2.6
and 2.8 respectively, they will be rounded to the same integer 3, yielding no change in the model
size for this layer when implementing quantization. In addition, in the whole search process, such
one integer choice may result in local convergence because the iteration process of one integer
quantization may get stuck in a local minimum region for the cursor search. To alleviate the above
two problems, we propose instead to make use of the nearest lower and upper integer bound at the
same time in the search training process. Compared to directly using the nearest one neighbor to
quantize, the lower and upper integer bounds may produce more variations in the loss function that
describes the quantization effects, yielding effective gradient changes to update the cursors more
efficiently. Subsequent experiments also demonstrate that this design can obtain better quantization
performance compared to simply applying rounding function on the searched cursor. As such, the
loss function for model size part in Eq.(2) is designed as follows:

$$Loss_C = (Cost(C))^\gamma \tag{3}$$

where $\gamma$ is a postive coefficient, $Cost(C)$ is a quantization related continuous cost function with
cursor $C$ as its variable. In this work and for the convenience of implementation, we further design
it as:

$$Loss_C = (\frac{\sum_{i=1}^N S_i \times c_i}{\sum_{i=1}^N S_i})^\gamma \tag{4}$$

where $S_i$ is defined as the size of the $i^{th}$ layer in bits when each of parameters is represented by 1
bit. In fact, for a trained model, the size of a layer in bits, i.e., $S_i$, is a constant for lay $i$. Since $c_i$ is
a continuous cursor, we may consider the above equation differentiable with respect to $c_i$.

When implementing the quantization for each layer, we utilize the DoReFa-Net (Zhou et al. (2016))
quantization:

$$w_k = 2Q_k(\frac{tanh(w)}{2\max(|tanh(w)|)} + 0.5) - 1 \tag{5}$$

where $w$ represents the full precision weight of a model and $Q_k(.)$ is the $k$-bit quantization function
that transforms a continuous value $x \in [0, 1]$ to a $k$-bit output $y \in [0, 1]$ as below:

$$y = \frac{round((2^k - 1) \times x)}{2^k - 1} \tag{6}$$

where *round* function is the typical rounding operation used in quantization. In other words, in
the process of quantization, after searching the possible quantization bit of $c_i$ in each layer, its

corresponding two nearest neighbor integers will be applied to Eq.(6) and Eq.(5) to quantize each layer of the network.

In the forward process of the proposed quantization scheme, output of the $i^{th}$ layer, assuming $O_i$, can be described with the following equation:

$$O_i = \sum_{j=1}^{M} ReLu(1 - |c_i - q_j|) \times O_j \quad s.t. \sum_{j=1}^{M} ReLu(1 - |c_i - q_j|) = 1 \qquad (7)$$

where $ReLu(x) = \max(0, x)$ is the Rectified Linear Unit that zeros out the negative values, $c_i$ is the cursor in one layer, $q_j$ denotes the $j^{th}$ quantization bit width within $M$ total quantization choices, and $O_j$ is the corresponding output of one convolution layer of the model when quantizing with $q_j$. Due to the introduction of $ReLu(x)$ in the forward process, the whole loss function may not be differetiable at the point of zero. In such a corner case, fortunately, the gradient may be still also estimated through the method of straight-through-estimator(STE) (Guo (2018)), whose details are beyond the scope of this paper. In this work, we mainly consider $q_j \in \{1, 2, 3, 4, 5, 6, 7, 8\}$, $M = 8$ and $c_i \in [1, 8]$, because it has been observed that with greater than 8 bits, the neural network's performance almost has no degradation (Elthakeb et al. (2018)). We also concentrate on such a design in all our experiments. In fact, there is also a new trend that investigates the possible quantization with bit that is not power of 2 (Elthakeb et al. (2018); Park et al. (2018b;a); Wang et al. (2018)). In addition, some hardware such as FPGA also gradually supports efficient quantization using such bits (Wang et al. (2018); Wei et al. (2019)).

Based on Eq.(7), it can be found that there is only two closest integers that take effects in each layer's output. As such, our proposed search algorithm can much reduce the coupling of different quantization operations, leading to more effective search of a good cursor. Assuming the cursor's lower and upper bound integer in the $i^{th}$ layer is $a_{i1}$ and $a_{i2}$, we can define two coefficients $d_{i1}$ and $d_{i2}$ as below:

$$d_{i1} = 1 - |c_i - a_{i1}|; \quad d_{i2} = 1 - |a_{i2} - c_i| \qquad (8)$$

where $c_i$ represents a cursor searched in the $i^{th}$ layer, $d_{i1}$ and $d_{i2}$ represents how close the bounds are to a cursor. The closer it is, the larger the coefficient is. A continuous move of a cursor then adjusts the weights continuously. Then, based on Eq.(7), given the input $X$ of one convolution layer, output of it (we only consider the quantization of the convolution layers in this work ) in the forward process of quantization can be simplified and described with the following equation:

$$O_i = d_{i1} \times (Conv(X, W_{i1}) + d_{i2} \times Conv(X, W_{i2})) \qquad (9)$$

where $W_{i1}$ and $W_{i2}$ are the temporary weights in one layer after quantization using $a_{i1}$ and $a_{i2}$ based on its corresponding 32 bit weight, $Conv$ is the convolution operation. The 32 bit weight will be updated as the whole algorithm iterates, while $W_{i1}$ and $W_{i2}$ will be recalculated in the forward process based on the new $a_{i1}$ and $a_{i2}$ . In other words, we activate the two closest bits to the cursor, and sum the convolution results of these two quantization bit choices based on the coefficients of the L1 distance. As such, the whole process is differentiable. This can be also intuitively explained by that the outcome of the desired quantization scheme for each layer in the forward process might be represented by a weighted sum of the two different quantization schemes using the approximated closest bit choices. Hence, the proposed scheme may find the best quantization scheme for the whole network in the cursor searching process based on the alternative optimization solution.

After the approximate alternative optimization approach converges or reaches the target epoch number, the final quantization bit in each layer can be obtained by applying rounding operation on each cursor for inference. The final quantized model may also need to be finetuned based on the quantization bit.

## 4 EXPERIMENTS

Currently, we only apply quantization on the weights and use full precision activations. In addition, we also follow the traditions in the domain of DNN quantization to avoid the quantization of the first and last layer in a model. In all the experiments, we take ResNet18, ResNet20, ResNet56 (He et al. (2015)) or MobileNetV2 (Sandler et al. (2018)) as the backbone models. Please be noted that these

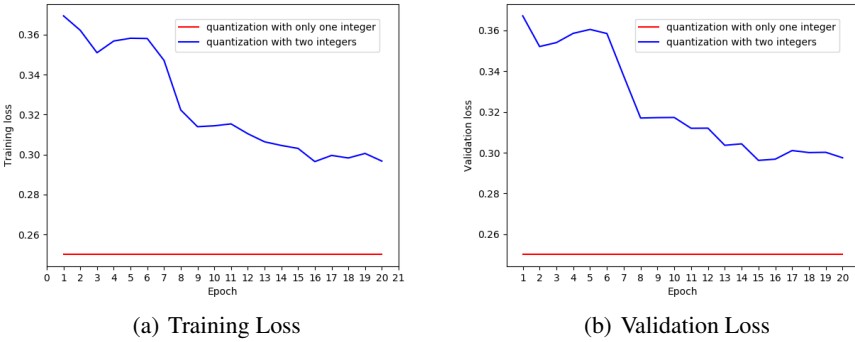

(a) Training Loss          (b) Validation Loss

Figure 1: Model size loss change for different quantization scheme.

models should be pretrained to obtain the floating point models first. For the initialization of cursors in each layer, all of them are set with 4 bits for the convenience of iteration. When the cursors are obtained by our method, the model may be further fine tuned to get its final accuracy, which is a practical tradition in the fields of NAS and quantization.

As for the parameter $\lambda$ in Eq.(2) and $\gamma$ in the loss of model size in Eq.(3), a rather optimal set of them is chosen as (0.25, 0.3) after trials. Based on our experiments, we also generalize a trial rule for $\lambda$. We control the balance of the two loss parts in Eq.(2) to ensure the ratio of the first and second part between [0.5, 2.0]. We also study the influence of $\lambda$ in the experiments to show that in most cases, the cursor based adaptive quantization scheme is not senstive to its change if $\lambda$ is at a larger interval of $\lambda \geq 0.1$. Concerning the learning rate schedule of weight and cursor, we apply cosine annealing method to adjust them. The minimum learning rate for them is 0.001 and 0.0001 respectively.

### 4.1 COMPARISON OF MODEL SIZE LOSS

To show the validity of quantization approach using two integer bounds nearest to the cursor, we first implement the search process by comparing it to using only one nearest integer of the cursor. We analyze their model size losses, i.e., Eq.(3), to show the great distinction in the training process.

Here we apply ResNet20 on CIFAR10 dataset to demonstrate the optimization process. For illustrative purpose, we only draw the training and validation loss change of the model size, i.e., the second term of Eq.(2), in the first 20 epochs. As shown in Figure 1, the red curve represents the training and validation loss of model size using one nearest integer to implement quantization, while the blue one denotes the training and validation loss of size obtained by using two neighbor integers nearest to the curser searched by the proposed scheme. The major differences in these two tests lie at the quantization choices. In fact, we also tried some other parameters and random initialization for one integer quantization scheme, and similar curves can be found. Obviously, the blue one looks more smooth and natural for a convergence process. The red loss may lead to a strong possibility that the cursors are stuck in a local minimum region instead. From this Figure, we can clearly notice that the training and validation loss of the model size using only one integer quantization will keep constant immediately in the first epoch, which is NOT desired for cursor search because it will cause no change in the model size anymore. In fact, the cursor values obtained by the one neighbor scheme tend to be 1 bit for all layers. The reason why the one integer quantization scheme fails may be because, in most cases, the weights in one layer span a rather small range, one lower integer quantization may lead to the same quantization results on the weights in the training process. Such same quantization results further yield almost no change in the backward gradient process, which is not beneficial for the optimal cursor search. The designed two integers' quantization process, on the other hand, can map the cursor to two different integer values, leading to efficient change in the model size even for the weights in rather a small value range. Figure 1 also shows that Eq.(4) is continuous and differntiable.

## 4.2 SEARCH PROCESS ANALYSIS

To get some insights of our adaptive cursor search algorithm, we investigate its iteration process in this subsection. For illustration only, we take MobileNetV2 on CIFAR10 as an example. Its search process is depicted in Figure 2 with the quantization bits ignored due to space limitation. Here the abscissa and vertical coordinate respectively represents the compression ratio and prediction accuracy. It should be noted that here our proposed algorithm runs 10 epochs only to clearly show the variation of performance. In addition, because of the cosine annealing scheduler, such an iteration process may also be representative. From Figure 2, we observe that for the proposed adaptive cursor search scheme, it first begins at the lower left region (lower accuracy and compression) and then gradually assembles to the upper right region (higher accuracy and compression). Meanwhile, there is some small vibrations in the whole process, for example, from epoch 8 to epoch 9, there is some increase in accuracy as well as compression ratio, but from epoch 9 to epoch 10, there is a slight reduction in both measures. It can also be noticed that the search process is rather stable and gathers to the final upper right region with better accuracy and compression ratio. We also observed similar pattern for ResNet20 and ResNet56 on CIFAR10, but we ignored the picture of them because of space limitation. The reason why the search process of our method can reach to a region with high prediction accuracy and compression ratio may be due to the alternative optimization approach to solve this bi-level problem with two goals. In addition, the regularization item may also play a positive role in this process.

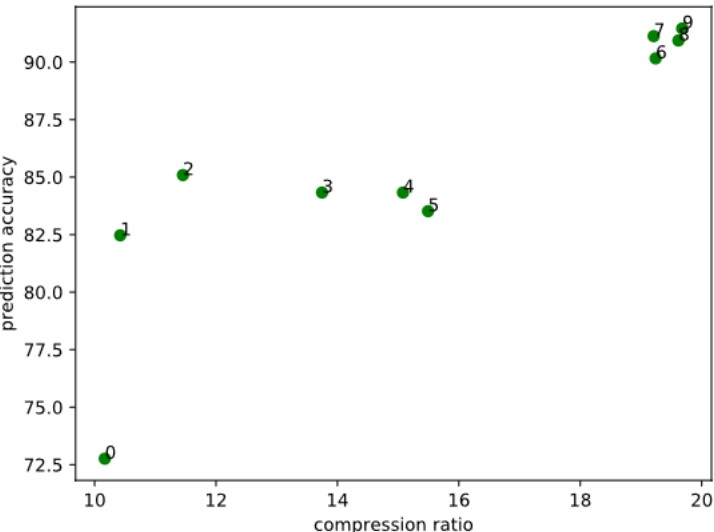

Figure 2: Adaptive search process for MobileNetV2 on CIFAR10, the number beside the points represent the epoch.

## 4.3 IMPACT OF REGULARIZATION COEFFICIENT

The regularization coefficient $\lambda$ in Eq.(2) determines the balance between the model precision and size. In this part, we carry out some experiments to analyze the influence of it on the whole performance. We choose $\lambda = 0.9, 0.7, 0.5, 0.25, 0.1, 0.05, 0.01$, and we test its effects on the quantized model. For the purpose of illustration, we test ResNet20 on CIFAR10. To directly show the effects of our cursor based differentiable search, we do NOT implement finetune step for all these results after finishing the cursor search. The results of the quantized ResNet20 on CIFAR10 is demonstrated in Table 1, and all the results are obtained by implementing the search with 200 epochs.

Table 1: Performance of ResNet20 on CIFAR10 with different $\lambda$

| $\lambda$ | 0.9 | 0.7 | 0.5 | 0.25 | 0.1 | 0.05 | 0.01 |
|---|---|---|---|---|---|---|---|
| Accuracy(%) | 90.10 | 90.14 | 90.18 | 91.79 | 91.03 | 91.16 | 91.58 |
| Compression ratio | 30.37 | 30.00 | 30.09 | 28.33 | 28.43 | 19.46 | 13.95 |

From Table 1, we can observe that for $\lambda >= 0.1$, the whole performance of the proposed quantization method is rather steady, that is, the accuracy and compression ratio of the quantized model maintain at a concentrated region with the accuracy about 90% while the compression ratio about 29.00. When $\lambda < 0.1$, the cursor based adaptive quantization approach may still have a good performance of prediction but gradually loses its effects on model compression. This can be explained that when the regularization becomes gradually weak, it does NOT exert its compression effects so well as when the coefficient is large. This further validates the effectiveness of the regularization function proposed in this paper.

## 4.4 CIFAR10 RESULTS

We demonstrate our cursor based adaptive quantization algorithm on CIFAR10 benchmark dataset with ResNet20, ResNet56 an MobileNetV2. For ResNet20,we compare the accuracy and compression ratio of the proposed approach to some related or similar works such as DNAS (Wu et al. (2018)), TTQ (Zhu et al. (2016)), PACT (Choi et al. (2018)) and LQE (Zhang et al. (2018)) with Resnet20 on CIFAR-10, and the details of accuracy and compression ratio are shown in Table 2. It can be noticed that, compared to the other related works, our method achieves much better compression ratio while achieving comparable or better classification accuracy on CIFAR10 dataset. The reason why the proposed approach is better than the quantization methods such as LQE, TTQ and PACT may be due to the adaptive cursor based search mechanism. By considering both the model accuracy and compression ratio, the cursor based approach can effectively search different quantization bit for each layer as a whole, leading to better compression ratio with better accuracy. Compared to DNAS, the reason for our better performance in terms of CR is partially due to that the two closest integers' quantization scheme produces less quantization error in each layer. In addition, it may be also because of our multiple lower bits' design in the search process.

Table 2: Performance comparison with other works using ResNet20 on CIFAR10

| | Accuracy(%) | Compression Ratio |
|---|---|---|
| Ours | 92.27 | 28.3 |
| DNAS(most efficient) | 92.00 | 16.6 |
| DNAS(most accurate) | 92.72 | 11.6 |
| LQE(2 bit) | 91.80 | 16.0 |
| TTQ(2 bit) | 91.13 | 16.0 |
| PACT(2 bit) | 89.70 | 16.0 |
| Baseline(32 bit) | 92.06 | 1.0 |

We also apply the proposed approach to ResNet56 and compare its performance with DNAS (Wu et al. (2018)), and the results are recorded in Table 5. We further test the proposed approach on MobilenetV2, with the results shown in Table 6. Because of space limitation, we put the detailed Tables and descriptions in Appendix.

## 4.5 CIFAR100 RESULTS

To further show the effectiveness of the proposed scheme, we test our method on CIFAR100 dataset using ResNet20, ResNet56 and MobileNetV2. We illustrate compressed ResNet20's performance compared to the original one on CIFAR100 in Table 3, it should be pointed out that here we also do not fine tune the original model, so its accuracy may not be the best one in the literature. For ResNet20, our approach achieves a good compression ratio of 11.6 while maintaining a comparable accuracy of 68.18%.

Table 3: Performance of ResNet 20 on CIFAR100

|                 | Accuracy | Compression Ratio |
|-----------------|----------|-------------------|
| Ours            | 68.18    | 11.6              |
| Original(32 bit)| 68.30    | 1.0               |

The performances of the quantized network of ResNet56 and MobileNetV2 on CIFAR100 datasets are presented in Table 7 and in Table 8 respectively, which are also placed in Appendix due to limited space. We notice that both quantized models show a little better accuracy with impressive compression ratios of 17.2 and 12.9 for ResNet56 and MobileNetV2 respectively.

## 4.6 IMAGENET RESULTS

In this subsection, we apply ResNet18 and MobileNetV2 to ImageNet dataset, which is a much larger dataset compared to CIFAR10 and CIFAR100. Here, as in (Han et al. (2015); Wang et al. (2018)), we present two sets of our results, i.e., the most efficient result and most accurate one to compare more conveniently.

We record the performance of the proposed method with ResNet18 on ImageNet in Table 4 as well as some comparisons to LQE (Zhang et al. (2018)), TTQ (Zhu et al. (2016)), PACT (Choi et al. (2018)) methods. From Table 4, it can be noticed that, compared to the original 32 bit model, the most accurate result of our algorithm achieves a promising compression rate of 13.9 with a slight accuracy drop of 0.15%, and for the most efficient one, our algorithm achieves an accuracy of 68.80% and an impressive compression ratio of 18.1. The most accurate result of our algorithm shows much better accuracy over LQE, TTQ and PACT methods although the compression ratio is a little bit smaller. As for the most efficient one, both the accuracy and compression ratio are better than those of LQE, TTQ and PACT, validating the effectiveness of the proposed scheme. The results of MobileNetV2 on ImageNet are illustrated in Appendix, please refer to Table 9 for details.

Table 4: Performance comparison with other works using ResNet18 on ImageNet

|                     | Accuracy(%) | Compression Ratio |
|---------------------|-------------|-------------------|
| Baseline(32 bit)    | 69.75       | 1.0               |
| Ours(most accurate) | 69.60       | 13.9              |
| Ours(most efficient)| 68.80       | 18.1              |
| LQE(2 bit)          | 68.00       | 16.0              |
| TTQ(2 bit)          | 66.60       | 16.0              |
| PACT(2 bit)         | 64.40       | 16.0              |

## 5 CONCLUSIONS

In this paper, we have proposed a novel cursor based DAS algorithm for obtaining the mixed precision DNN model. Different from most of the traditional approaches, which choose quantization configuration using heuristics or learning based rules, we adaptively choose the quantization bit for each layer in the DNN model from the perspective of NAS. A cursor based search algorithm with alternative manner is applied for efficient optimization. The nearest two neighbor integers to the cursor are used to implement the quantization in the training process to reduce the quantization noise and avoid local convergence. The result of our algorithm is the adaptive bit width choice for different layers as a whole. Extensive experiments with some typical models demonstrate that the proposed approach provides dramatic compression capability with accuracy on par with or better than the state-of-the-art of methods on benchmark datasets. In the near future, we will apply the proposed scheme to object detection tasks to further show its possible wider application. We may also utilize some hardware platforms to test some performance such as inference time and resource consumption.

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

## A  APPENDIX

Due to space limitations, we put a number of experimental results here.

### A.1  RESNET56 ON CIFAR10

We apply the proposed approach to ResNet56 and compare its performance with DNAS  (Wu et al. (2018)), and the results are recorded in Table 5. In this Table, we notice that ResNet56 compressed using our quantization approach shows a higher compression ratio of 20.6 compared to the most efficient DNAS's result of 18.9, while its accuracy is almost the same. As for the most accurate one of DNAS, our quantized model is still comparable, while the compression ratio is much higher compared to 14.6. It should also be pointed out that due to different environments and other implementation differences in the whole process, the accuracy of our ResNet56 baseline is not as good as the one in DNAS. It should also be noted that DNAS takes cutout operation in its experiments, we also test such an operation to show the comparison. It may be noticed that our test with cutout achieves a accuracy of 94.48% with an impressive compression ratio of 23.3.

Table 5:  Performance comparison with other works using ResNet56 on CIFAR10

|  | Accuracy(%) | Compression Ratio |
|---|---|---|
| Ours | 93.45 | 20.6 |
| Ours + cutout | 94.48 | 23.3 |
| Baseline of our implementation(32 bit) | 93.39 | 1.0 |
| DNAS + cutout(most efficient) | 94.12 | 18.9 |
| DNAS + cutout(most accurate) | 94.57 | 14.6 |
| Baseline in DNAS(32 bit) | 94.42 | 1.0 |

### A.2  MOBILENETV2 ON CIFAR10

We further apply the proposed approach to MobilenetV2, which is a typical DL model for mobile devices and embedded systems, with the results shown in Table 6. It can be noticed that our adaptive cursor based quantization shows a better classification accuracy of 93.28% as well as a promising compression ratio of 12.4.

Table 6:  Performance of MobileNetV2 on CIFAR10

|  | Accuracy | Compression Ratio |
|---|---|---|
| Ours | 93.28 | 12.4 |
| Original(32 bit) | 92.39 | 1.0 |

### A.3  RESNET56 ON CIFAR100

We also test ResNet56 on CIFAR100 and present its corresponding results in Table 7. We can see that the proposed algorithm yields a little bit better accuracy(1.05%) together with an impressive compression ratio of 17.2.

Table 7:  Performance of ResNet56 on CIFAR100

|  | Accuracy | Compression Ratio |
|---|---|---|
| Ours | 71.84 | 17.2 |
| Original | 70.79 | 1.0 |

## A.4 MOBILENETV2 ON CIFAR100

The quantized MobilenetV2 on CIFAR100 and its corresponding results are demonstrated in Table 8. As for MobileNetV2, the proposed algorithm yields a little bit better accuracy together with an impressive compression ratio of 12.9.

Table 8: Performance of MobileNetV2 on CIFAR100

|  | Accuracy | Compression Ratio |
| --- | --- | --- |
| Ours | 68.04 | 12.9 |
| Original(32 bit) | 67.96 | 1.0 |

## A.5 MOBILENETV2 ON IMAGENET

The performance of MobileNetV2 on ImageNet is illustrated in Table 9 together with comparisons to some related works such as HAQ (Wang et al. (2018)) and deep compression (Han et al. (2015)). In Table 9, we notice that, for the most accurate result, the quantized MobileNetV2 model using our approach shows a slight accuracy loss (71.65% vs 72.19% of the original 32 bit model) while achieves an encouraging compression ratio of 9.1. It may also be observed that the accuracy of our most accurate one is a little bit higher than the corresponding most accurate results of HAQ and deep compression together with a better compression ratio. While for the most efficient one, our algorithm shows a compression ratio of 14.3, which is better than that of HAQ, but smaller than that of deep compression. However, our approach demonstrates a dramatically better accuracy of 70.59% compared to the corresponding 66.75% of HAQ and 58.07% of deep compression, which again validated the advantage of our algorithm.

Table 9: Performance comparison with other works using MobileNetV2 on ImageNet

|  | Accuracy(%) | Compression Ratio |
| --- | --- | --- |
| Baseline(32 bit) | 72.19 | 1.0 |
| Ours(most accurate) | 71.65 | 9.1 |
| Ours(most efficient) | 70.59 | 14.3 |
| HAQ(most accurate) | 71.47 | 7.5 |
| HAQ(most efficient) | 66.75 | 13.9 |
| Deep compression(most accurate) | 71.24 | 8.0 |
| Deep compression(most efficient) | 58.07 | 16.0 |

