# OpenReview forum: "CURSOR-BASED ADAPTIVE QUANTIZATION FOR DEEP NEURAL NETWORK"
_ICLR.cc/2020/Conference — Reject_

### Official Review · AnonReviewer3 · 2019-10-25
**Official Blind Review #3**

**Rating:** 6

**Review:**

The paper proposes a novel way to determine automatically the quantization level at each layer of
deep neural networks while training. The quantization can result in a more compact network requiring
less memory for storing the parameters (weights), with minimal drop in accuracy.
If I understand correctly, the main idea in the paper is utilizing a continuous representation of the quantization using a continuous cursor, and a new differential approach from Neural Architecture Search (NAS), to create a differential learning algorithm for the problem of finding the optimal quantization.
On the positive side, the authors present very good experimental results on popular networks like ResNet,
and improve on other method's compression accuracy, with little to no reduction in accuracy.
Howerver, the paper is not clearly and carefully enough written to convey the author's results which makes it a borderline paper in my opinion.


In my understanding, compressed (quantized) networks can have different advantages besides a more compact representation of the network:
For example, this can also reduce training time because all operations are performed with words with fewer bits - but the authors focus only on the compression ratio and do not discuss this issue. In fact, it seems that their algorithm actually requires more training time than standard training of neural networks, because you start with training w with full 32-bit representation, and then train with different quantizations simultaneously.
It would be good if the authors describe clearly how do different methods (theirs and the others mentioned) compare to each other in terms of savings of different resources (training time, compression, inference/prediction time for the trained networks etc.)  and which of these is more important (e.g. the latter in mobile devices).

Another practical issue is that using a different number of bits for each layer may complicate the design and software/hardware implementation of a network, compared to say allocating 4-bits or 8-bits to each weight. It is correct that you can reduce the overall number of bits, but you may reduce it even further if you allow a different number of bits for each individual weight. I assume that there is a price in power/memory/speed etc. to this non-homogenous property, and it may be better to get a slightly lower compression ratio provided that there is more uniformity in the network weights.  This is not a critique of this particular paper, but of the entire framework of flexible quantization.

The English of the manuscript could be greatly improved. For example:
Page 2: "..multiple bits for different layers.." - I assume the authors mean "..a different number of bits used for different layers .."
Page 2, bottom: "The authors (Wang et al. 2018) presented ..." - the sentence is long and unclear.
Many sentences starting with 'And' - for example: "And a Gumbel softmax function ..." (page 3)

There are a few vague and non-informative statements which do not contribute to the understanding of the field and the authors' contribution. For example:
- Page 3: "A typical search method is random search, however, its efficiency is not ideal." (does efficiency refer to computational efficiency here?)
- Page 4: "The reason why we add regularization item to the loss function is because the regularization can prevent overfitting to some extent" (I thought that here the main purpose of regularization is to get a trade-off between accuracy and compression).
- Page 6: Regarding parameters choice: 'a rather optimal set of them is chosen ..' - this seems quite arbitrary and it is not clear how to choose parameters for the authors' method for future architectures and datasets.


There are also terms that are known to expert but could use a short explanation/reference - for example:
- Adam optimizer
- Cosine annealing
- Gumble Softmax

The mathematical equations are definitions are not clear enough and contain errors:
- Eq. (1): there is (x',y') twice in the equation, but in the sentence afterward there is also (x,y).
There are D_T, D_V with and without tilde.
The space of maximization of w for a given quantization is not defined (I'm assuming all weights vectors w
which can be represented using the number of bits in the quantization).
- Eq. (3) - the loss is loosely defined - what is 'parameter size'? the number of bits for all the weights in each layer?
- Eq. (4) - is w or w_k the full precision weights? why are x and y in the interval [0,1]?
- Eq. (6) - c_i represnts the i-th layer, but d_1,d_2, a_1,a_2 are fixed. Are they also different for different layers?
Also, a_1 and a_2 are the boundaries of the cursor - are they set to [0,1]? or [0,32]?
- The authors defined Conv - the convolution operation - but what are W_1 and X? vectors? how is the convolution
defined precisely? it is also not common in math papers to use '*' for  product

It is not entirely clear to me how exactly the authors define a differentiable loss and a gradient for the quantization part Loss_q.  When a cursor parameter changes (e.g. c_i from 2.5 to 2.6) then the loss is defined by quantization to the two consecutive integers (in that case 2,3 ) and there is a continuous mixture parameter between them which can be changed smoothly, which is nice. But at some point, the parameter will reach 3, and then the quantization will use the two integers 3 and 4 - the fact that 3 has weight 1 in the loss (and 2,4 have weight zero) makes the loss continuous, but is the loss differentiable at this point?  some explanation is needed here on if this is a problem and how is it avoided/solved.

Algorithm 1: The description in Figure 1 can be made more precise. The algorithm seems to alternate between
optimizing the cursor c, and optimizing the weights w for a given value of c.
There are also missing details and parameters like gradient step size.  What does Grad-C * L_C mean? is the gradient multiplied by the loss?? or only applied to it?
In the sentence "Quantize the network ... to update the loss" - which loss is updated and how? L_V, L_T? something else?
A detailed description of the algorithm and parameters or the actual code used by the authors would improve the understanding of their method.

Section 4.1: Figure 2 compares the loss of one-integer vs. two-integers quantization scheme. The authors argue that their two integers scheme is better because it is smoother. But the loss for one integer is actually lower - so why wouldn't it be better to use this one?
will the two-integer method eventually reach a lower loss?



**Experience Assessment:**

I do not know much about this area.

**Review Assessment: Checking Correctness Of Derivations And Theory:**

I carefully checked the derivations and theory.

**Review Assessment: Checking Correctness Of Experiments:**

I carefully checked the experiments.

**Review Assessment: Thoroughness In Paper Reading:**

I read the paper thoroughly.

---

> ### Author Response · Authors · 2019-11-15
> **We improved the presentation of the whole manuscript to make it more clear**
>
> We really appreciate the detailed comments of the reviewer.
> Reply to "However, the paper ...":
> We improved a lot about the presentation, which we believe to make the whole manuscript more clear now.
>
> Reply to "In my understanding, ...":
> Yes, the proposed scheme needs more time for training. The above points about resources and so on are worth more comprehensive study, however,  it is not the focus of this work. Due to hardware resource related problems, currently, we may not carry out these related tests. We included this point as part of our future work as shown in the revised manuscript.
>
> Reply to "Another  ...":
> Yes, the comments are right. It is the state-of-the-art, in fact, non-homogeneous quantization may bring extra cost in power. Fortunately, recently some hardware platforms emerged to support this kind of operations such as FPGAs and ASICs.
>
> The English problems:
> Page 2: "..multiple bits for .."
> Answer:  Yes, we revised the wordings as suggested.
>
> Page 2, bottom: "The authors (Wang et al. 2018) presented ..."
> Answer:  Yes, we rewrote the sentence to better convey our meanings.
>
> Many sentences starting with 'And' (page 3)
> Answer:  We improved all such expressions in the revised manuscript.
>
> - Page 3: "A typical search ..."
> Answer:  Thanks for pointing out, we clarified it by adding a word “computational”.
>
> - Page 4: "The reason  ....".
> Answer:  Yes, we supplemented this point before the previous words, thanks.
>
>  - Page 6: "Regarding parameters choice: .... "
> Answer:  We supplemented some words on this aspect to guide the parameter choice. Please refer to the middle part in page 7 of the revised manuscript.
>
> "There are... "
> Answer:  Thanks a lot for the comments, due to the space limitation, we ignored these detailed explanations.
>
> The mathematical .....
> Answer:  Sorry for the typos in the equation, we corrected them in the revised manuscript.
>
> - Eq. (3) - the loss ...
> Answer:  We updated this equation and formally define the layer size as $S_{i}$ in the revised manuscript.
>
> - Eq. (4) - is w or w_k ...
> Answer:  In Eq.(4), $w$ is the original full precision weights, and $w_k$ is the quantized weight. $x$ and $y$ are constrained to the interval $[0, 1]$ for the convenience of round operation, and there is a scale factor $1/(2^k – 1)$ as shown in Eq.(5).
>
>  - Eq. (6) - c_i ...,
> Answer:  Yes, they are different for different layers. We rewrote them to clear this confusion.
>
> - Also, a_1 and a_2 ...?
> Answer:  In this paper, $a_{i1}$ and $a_{i2}$ are confined to $[1, 8]$ since as mentioned explicitly in the manuscript, we only consider the lower bit quantization (1, 2,3,4,5,6,7,8 bit) in this work.
>
> - The authors defined Conv ...
> Answer:  As pointed out in the manuscript, $W_{i1}$ is the weight quantized with the lower bound $a_{i1}$ for one layer, and $W_{i2}$ is the weight quantized with the upper bound $a_{i2}$ for one layer. $X$ is the input to this layer.
>
>  It is not entirely clear ...
> Answer:  Good point! Thanks for the insightful comments. For this case, based on our forward quantization equation(7) as shown below,
> $$O_{i}= \sum_{j=1}^{M} ReLu(1 - |c_{i} - q_{j}|)\times O_{j}  \quad   s.t. \sum_{j=1}^{M} ReLu(1 - |c_{i} - q_{j}|)=1 $$
> This is ignored in the previous manuscript and now supplemented with some details. Only 3 bit will be used for quantization in this process. Due to the existence of ReLu function, which is not differentiable at point zero, so the whole loss function may not be differentiable anymore. At these points when the loss function is not differentiable, we can apply a typical approximation techniques such as straight-through-estimator(STE)[5], to make the loss function differentiable again. We added some words in the revised manuscript as suggested, please refer to page 6 in the revised manuscript.
>
> [5]Yunhui Guo. A survey on methods and theories of quantized neural networks. CoRR,
> abs/1808.04752, 2018. URL http://arxiv.org/abs/1808.04752
>
> Algorithm 1: The description ...
> Answer:  Sorry for the equation edition problem, we corrected them and made them more clear. In addition, we also added some words to explain the whole algorithm. Please refer to page 4 and the new flowchart figure in Page 5.
>
> In the sentence "Quantize ...”.
> Answer:  We supplemented some words on this point to make it more clear. We also corrected the typos in the flowchart. Please refer to page 4 and page 5 about this part. We will release our codes upon acceptance.
>
> Section 4.1: Figure 2  ... ?
> Answer:  Although the training loss is smaller, but it will converge quickly to almost a constant value, leading to no gradient change anymore, and the algorithms output will be all 1 bit. That is NOT we need in fact. Instead, we need a rather smooth convergent process that leads to better trade-off between model size and accuracy.  Based on our previous experiments, the two bits’ quantization scheme can reach comparable loss of one bit quantization after enough number of epochs.

---

### Official Review · AnonReviewer2 · 2019-10-28
**Official Blind Review #1**

**Rating:** 3

**Review:**

This paper is about using quantization to compress the DNN models. The main idea is to use NAS to obtain the mixed precision model. More specifically, it adaptively chooses the number of quantization bit for each layer using NAS by minimizing the cross-entropy loss and the total number of bits (or model size) used to compress the model. The experiment is on CIFAR and ImageNet, and compared with other quantization methods showing better accuracy.

I have somes questions on this paper:

1) Is Eq(1) standard for quantization optimization? Any reference? Normally, we aim to minimize the loss on the training set, not the validation set,  or sometimes generalization loss on data from the data distribution.

2) For experiment, it is interesting to see how the compression ratio changes over the accuracy--- that is a curve with x-axis on compression ratio, and y axis  accuracy so that we can have a sense of how the accuracy and compression rate trade-offed.

3) What is the training time for NAS for finding the 'optimal' bits for each layer? Although we might not care much about the training time for compression task, I just want to have a sense of how training works. Also what is the reason for not compressing the first and last layers? Do these two layers taken into account in the final compression ratio computation?

**Experience Assessment:**

I have published in this field for several years.

**Review Assessment: Checking Correctness Of Derivations And Theory:**

I assessed the sensibility of the derivations and theory.

**Review Assessment: Checking Correctness Of Experiments:**

I assessed the sensibility of the experiments.

**Review Assessment: Thoroughness In Paper Reading:**

I read the paper thoroughly.

---

> ### Author Response · Authors · 2019-11-15
> **We provided detailed answers to the reviewer's questions, and we added another 4 sets of tests to show the performance of the proposed scheme**
>
> Thanks a lot for the reviewer's suggestions and comments,  we replied carefully one by one:
>
> Reply for question (1):
> Thank you for your comments. Eq.(1) is a rather standard process for NAS, which is different from other typical problems in the field of deep learning. As suggested, we added one reference [1] in the revision. Here, we use training data for the weights update, while the validation data for the update of the architecture parameter. But it is a little bit different due to the introduction of $cursor$ in our problem definition. The differences of our cursor based adaptive search with the other related approaches such as DARTS and DNAS is emphasized in the manuscript, which can be found at the bottom of Page 4.
>
> Reply for question (2):
> Thanks for the comments. In Table 1, we have shown the influence of the regularization parameter on both the accuracy and compression ratio. It may be noticed that when $\lambda \geq 0.1$, the proposed scheme can achieve a stable good performance of both compression ratio and accuracy. When $\lambda < 0.1$ the cursor based adaptive quantization approach may still have a good performance of prediction but gradually loses its effects on reducing model size. In this range, it can be also observed that there seems to be a trade-off between the compression ratio and accuracy.
>
> Our search procedure is an interesting process to analyze. As shown in Fig.2, we have drawn a curve that shows the search process, illustrating the accuracy and compression ratio changes in the whole process. The search process starts from the bottom left of the accuracy vs. compression ratio plane, and can gradually ascend to the upper right corner.  We attribute this expected good behavior to the following several reasons. First, we used alternative optimization approach to solve this bi-level problem with two goals. The bi-level problem refers to the optimization of weights and cursor at the same time. In addition, the regularization item may also play a positive role in this process. Last but not the least, we make use of cosine annealing method to adjust the learning rate from 0.1 to 0.001. This learning rate schedule function can efficiently avoid the local minimum value in the objective function during the search process.
>
> Due to space limitations, we did not add more experiments to draw this curve. Instead, we added another 4 sets of experiments using new backbone network ResNet 56 and new dataset CIFAR100 as suggested by the other two reviewers to show the effectiveness of the proposed scheme.
>
> Reply for question (3):
> Thanks for your comments. Actually the training time is NOT the focus of this work.  To provide the reviewer a sense on this aspect, we further pointed out some facts as follows: On our P40 GPU machine, our training time for ResNet 20 on CIFAR10 is about 10  hours. As for ResNet56 and MobileNetV2, the training time on CIFAR10 for both networks is about 20 hours.
>
> Reply for the training process works as follows:
> With the initialized cursors, we sample the training data, quantize the weights of each layer in the network with the two closet integers to the cursor, calculate the loss function, and update the weights of each layer in the network using gradient descent algorithm. Then, sample the validation data and also quantize weights with the two closet integers, and calculate the loss function, update the cursor values with gradient descent. Such a procedure iterates until it converges or reach the target number of epochs. So it is an alternative optimization training process.
>
> Reply for not quantizing the first and the last layer:
> This is a practical convention in the field of quantization for DNNs since these two layers severely affect the performance of the network. This is also the case in the reference of DNAS[1], PACT[2], LQE[3], TTQ[4], and so on. Since we do notquantize them, they are not taken into account in the final compression ratio calculation. This is also a convention in the field.
>
> [1] Bichen Wu, Yanghan Wang, Peizhao Zhang, Yuandong Tian, Peter Vajda, and Kurt Keutzer.
> Mixed precision quantization of convnets via differentiable neural architecture search. CoRR,
> abs/1812.00090, 2018. URL http://arxiv.org/abs/1812.00090.
> [2] Jungwook Choi, Zhuo Wang, Swagath Venkataramani, Pierce I-Jen Chuang, Vijayalakshmi Srinivasan,
> and Kailash Gopalakrishnan. PACT: parameterized clipping activation for quantized neural
> networks. CoRR, abs/1805.06085, 2018. URL http://arxiv.org/abs/1805.06085.
> [3] Dongqing Zhang, Jiaolong Yang, Dongqiangzi Ye, and Gang Hua. Lq-nets: Learned quantization
> for highly accurate and compact deep neural networks. CoRR, abs/1807.10029, 2018. URL
> http://arxiv.org/abs/1807.10029.
> [4] Chenzhuo Zhu, Song Han, Huizi Mao, and William J. Dally. Trained ternary quantization. CoRR,
> abs/1612.01064, 2016. URL http://arxiv.org/abs/1612.01064.

---

### Official Review · AnonReviewer1 · 2019-10-30
**Official Blind Review #1**

**Rating:** 3

**Review:**

The authors developed a novel quantization technique that yields layer-wise different mixed-precision quantization. To do so, they alternatively update the pre-trained weights and the quantizer, which they call cursor. The following two features distinguish this paper: using two precision values around the cursor's value (instead of the closest one) and regularizing the parameter size using a new loss function. Because the whole process is differentiable, the appropriate precision to each layer can be found fast. Thanks to these efforts, this method balances the compression rate and accuracy well on CIFAR-10 and ImageNet.

Unfortunately, the details of these two features are hardly supported in conceptual, theoretical, and experimental manners.
(1) The reviewer guesses that "parameter size after quantization in one layer" in equation 3 is related to "layer_size" in equation 8. However, the relationship is unclear and cannot convince the reviewer that the newly proposed loss (equation 3) is differentiable. Also, equation 7 ($f=d_1*(Conv(X, W_1)+d_2*Conv(X*W_2))$) is difficult to make the reviewer understand why and how this method works.
(2) The preliminary experiment in section 4.1 seems to claim that a single cursor leads a network to local minima, by only showing training loss. The reviewer thinks that the authors need to show validation loss as well to claim the failure of the single cursor.


The followings are minor comments.
* To the best of the reviewer's knowledge, the term "cursor" is the authors' original one. Therefore, the authors need to write its definition.
 * The mathematical notations in this paper are confusing. (1) Each face has different meanings. For example, $max$ means $m \times a \times x$, rather than the max operator $\max$. (2) operator * is used in arbitral as both unitary and binary without any comments or definitions.
* The reference is required to be format and cited correctly. For example, [He et al. 2015] is accepted to CVPR 2016 but is not mentioned.
* The authors claim that "comprehensive experiments" are done. However, the authors' proposal is experimented only on CIFAR-10 and ImageNet with MobileNet V2 and ResNet-18, while DNAS, for example, is verified on more variety of data and networks, plus object detection tasks. The reviewer thinks that the method proposed in this paper requires more comprehensive experiments.


**Experience Assessment:**

I have published one or two papers in this area.

**Review Assessment: Checking Correctness Of Derivations And Theory:**

I carefully checked the derivations and theory.

**Review Assessment: Checking Correctness Of Experiments:**

I assessed the sensibility of the experiments.

**Review Assessment: Thoroughness In Paper Reading:**

I read the paper at least twice and used my best judgement in assessing the paper.

---

> ### Author Response · Authors · 2019-11-15
> **We rewrote section 3.4 to better explain why the loss function and the cursor search process is differentiable, we also added another 4 sets of new experiments to show the performance of the proposed scheme.**
>
> Many thanks for the reviewer's insightful comments, and please refer to our responses below:
>
> Reply for point (1):
> We apologize for the previous careless edition of the equations you mentioned above. We corrected the confusing points in Eq.(7) (now Eq.(9)) and rewrote Eq.(3). We carefully revised section 3.4 to better explain why the proposed loss function is differentiable. Please refer to Eq.(4) in the revised paper:
> $$Loss_{C}=(\frac{ \sum_{i=1}^{N}S_{i}\times c_{i}}{\sum_{i=1}^{N}S_{i}})^{\gamma} $$
> where $S_i$ is the size of the $i^{th}$ layer in bits when each of parameters is represented by 1 bit, $c_i$ is the continuous variable that represents the cursor of the $i^{th}$ layer, $\gamma\in\mathbb{R}^+$ is a fixed coefficient. For a trained model, the $S_i$ for layer $i$ will be a constant. As such, the loss function is continuous and differentiable. We hope this equation clears the doubt why the loss function is differentiable.
>
> Previous Eq.(7) (now Eq.(9) in the revised manuscript) describes the forward process of our quantization strategy. The whole process can be summarized as follows:
> With the training and validation set, initialized cursor value and a pretrained model as the inputs, our algorithm first quantizes weights in each layer of the network with the two most close integers to the cursor and calculates the loss based on the training data, and then it updates the original 32 bit weights by using gradient descent algorithm.  For the subsequent validation, it also first quantizes weights in each layer of the network with the two most close integers that are within the distance of 1 to the cursor, and use them to obtain the validation error, then our algorithm utilizes this error to update the cursors with gradient descend algorithm. It should be emphasized that we utilize the original 32 bit weight for sharing when implementing quantization with the cursor’s two neighbor integers.
>
> Specifically, please refer to Eq.(9) in the revised paper, which describes the forward process of the proposed quantization scheme:
>  $$      O_{i} = d_{i1}\times (Conv(X,W_{i1} )+d_{i2}\times Conv(X,W_{i2} ))   $$
> where $X$ is the input tensor, $W_{i1}$ and $W_{i2}$ are the temporary weights in one layer after quantization using two neighbor integers to the cursor based on their corresponding 32 bit weights,  $Conv$ is the convolution operation, $d_{i1}$ and $d_{i2}$ are the distances to the cursor. The 32 bit weights will be updated within the iterations in the algorithm, while  $W_{i1}$ and $W_{i2}$ will be recalculated in the forward process based on the updated two neighbor integers to the cursor. In other words, we activate the two closest bits to the cursor, and sum the convolution results of these two quantization bit choices based on the coefficients of the L1 distance.
>
> This can be also intuitively explained by the fact that the outcome of the desired quantization scheme for each layer may be represented by a weighted sum of the two different quantization schemes using the approximated closest two bits. Hence, the proposed scheme may find the best quantization scheme for the whole network in the cursor searching process based on the alternative optimization solution.
>
> Reply for (2):
> Thank you for your comments, we have added a figure for the validation loss in Figure 1 to show that quantization using one integer fails here.
>
> For minor comments:
> * 1 ) Answer:  Thanks for the kind suggestion, we defined the cursor in the revised paper. Please refer to the bottom of Page 3. It reads “If  we  further  consider the  possible  bit  for  each  layer  as  a  virtual  continuous  cursor  in  the  range  of $[1, 32]$,  the  cursors then become significant parts of the architecture for a neural network model. Here, we define the $cursor$ as a position that is related to the quantization choice for each layer. Its value is a floating-point number within $[1, 32]$. ”
>
> * 2) Answer:   Thanks for the comments, we corrected inconsistent notations and carefully chose mathematical symbols in the equations as suggested.
>
> * 3) Answer:   Thanks for the comments, we reformatted this reference.
>
> * 4) Answer:   Thanks for the suggestions, we added another 4 sets of experiments, including ResNet56 on CIFAR10, ResNet56 on CIFAR100, ResNet20 on CIFAR100, MobileNetV2 on CIFAR100, to further validate our theoretical contributions. Due to page length limitation, we put these additional results in Appendix.  We also noticed that there's no object detection experiments in the DNAS paper, but we agree with the reviewer that more experiments besides image classification such as object detection can further validate our novel adaptive quantization scheme, which is subject to our future work.

---

### Author Response · Authors · 2019-11-15
**We carefully revised the manuscript based on the comments.**

Dear reviewers and ACs,

We have revised our manuscripts according to reviewers’ comments and addressed all concerns. We sincerely thank reviewers for their detailed and insightful comments. We are happy to see our manuscripts is significantly improved and we would be very grateful if we are informed about any further concerns.

To summarize our key changes:

1.	We rewrote section 3.4 to better explain why the loss function and the cursor search process is differentiable. We hope this answered the questions raised by reviewer 1 and reviewer 3.

2.	We added more experiments to further validate our proposed adaptive quantization scheme. Specifically, we added 4 sets of experiments, including ResNet56 on CIFAR10, ResNet56 on CIFAR100, ResNet20 on CIFAR100, and MobileNetV2 on CIFAR100. All experimental results achieved decent accuracy and compression ratio and outperform baseline and existing work. We hope this could improve our paper’s overall results and better convince all three reviewers about the advantages of our quantization scheme.

3.	We corrected a few typos in previous Equations, and now all the representations in the Equations are consistent.

4.    We improved the presentation in details based on all reviewers' comments one by one.

Kind regards,
Authors

---

### Decision · Program_Chairs · 2019-12-19

**Decision:**

Reject

**Comment:**

This paper presents a method to compress DNNs by quantization. The core idea is to use NAS techniques to adaptively set quantization bits at each layer. The proposed method is shown to achieved good results on the standard benchmarks.
Through our final discussion, one reviewer agreed to raise the score from ‘Reject’ to ‘Weak Reject’,  but still on negative side. Another reviewer was not satisfied with the author’s rebuttal, particularly regarding the appropriateness of training strategy and evaluation. Moreover, as reviewers pointed out, there were so many unclear writings and explanations in the original manuscript. Although we admit that authors made great effort to address the comments, the revision seems too major and need to go through another complete peer reviewing. As there was no strong opinion to push this paper, I’d like to recommend rejection.